# Effect of Benzotriazole on the Localized Corrosion of Copper Covered with Carbonaceous Residue

**DOI:** 10.3390/ma14112722

**Published:** 2021-05-21

**Authors:** Yun-Ho Lee, Min-Sung Hong, Sang-Jin Ko, Jung-Gu Kim

**Affiliations:** School of Advanced Materials Science and Engineering, Sungkyunkwan University (SKKU), Suwon 16419, Korea; yunho0228@naver.com (Y.-H.L.); smith803@naver.com (M.-S.H.); tkdwls1315@naver.com (S.-J.K.)

**Keywords:** copper corrosion, localized corrosion, BTA inhibitor, inhibition, carbonaceous residue

## Abstract

Carbonaceous residues on copper pipes during the manufacturing process are known to be one of the main causes of pitting corrosion on copper pipes. This study examined the corrosion-inhibiting effect of benzotriazole (BTA) on C12200 copper pipes with carbonaceous film in synthetic tap water. In the absence of BTA, localized corrosion mechanisms due to galvanic corrosion, crevice corrosion, and oxygen-concentration cell were proposed in the boundary part of the carbonaceous film on the copper through X-ray photoelectron spectroscopy (XPS), scanning electron microscopy (SEM) with energy dispersive spectrometer (EDS) analyses. Electrochemical tests showed that BTA inhibits corrosion by forming Cu−BTA complexes on all over the copper surface where carbonaceous film is present. BTA mitigates galvanic corrosion and crevice corrosion at the boundary of the carbonaceous film and suppresses the formation of oxygen-concentration cells through the formation of a Cu−BTA complex.

## 1. Introduction

Copper has excellent corrosion resistance and has been widely used for the tube of pipe-borne water. More than 80% of all water pipes in Europe and North America are made of copper [1]. However, despite the excellent corrosion resistance of copper, leakage problems due to pitting corrosion occur constantly. Extensive research has been conducted in this regard in recent years. Many research papers have been reported, including excess carbonaceous manufacturing residues on pipe surfaces, soldering flux, water chemistry, microbial activity, and other variables within a given water distribution system [2,3,4,5,6].

The most widely used method for preventing copper corrosion is the addition of corrosion inhibitors. Benzotriazole (BTA) is one of the most effective corrosion inhibitors for copper, and many studies have been conducted on BTA. Figure 1 shows the chemical structure of BTA. When copper is immersed in a solution containing BTA, it is believed that BTA forms a protective layer in the solutions [7]. The corrosion-inhibiting mechanism of BTA is generally accepted as the formation of a protective layer on the Cu surface by the formation of a Cu−BTA complex by a Cu−N bonds in BTA [8,9]. Although there is controversy regarding BTA inhibiting mechanisms, many researchers have studied BTA as an effective inhibitor for different conditions. Walker showed that inhibiting efficiency (*IE**%*) was 100% when 1 mM BTA was added under dilute seawater solution, dilute NaNO_2_ solution, and dilute NaCl solution conditions [10]. Musiani showed that BTA was less effective at a low pH level [11]. P. Yu researched the inhibition efficiency of BTA on copper in deionized water. When BTA concentration exceeded 8 ppm in deionized water, *IE%* exceeded 80% [12]. Ross and Berry investigated BTA inhibitory effectiveness under 10% H_2_SO_4_, low or high flow rate solution [13]. In addition, many studies on the inhibition efficiency of BTA under various conditions were conducted [7,9,14,15,16].

However, most of the experimental conditions mentioned above are different from the actual pipe conditions in use because these experiments were performed on a polished copper surface. Copper pipes used in fields contain carbonaceous residues from the manufacturing process. Carbonaceous residues are produced by the oxidation of the drawing oil on the copper surface during pipe drawing and soft annealing. Carbonaceous residues are known to be the major cause of pitting corrosion of copper pipes [2,17,18]. To prevent this type of corrosion, the UK and other European countries provided a standard specification on the amount of carbon content on pipe surfaces after cleaning copper pipes during the manufacturing process [19]. However, in most countries, such as the Republic of Korea, that do not have a standard specification, carbonaceous residues are found on copper pipes after the manufacturing process. In recent years, corrosion due to carbonaceous residues has been identified as a serious problem. A large amount of pitting corrosion occurred within 3–4 years after use in the sprinkler copper tube. As a result of analysis of the leaking tube, it was confirmed that carbonaceous residues were continuously distributed in a thickness of 20–30 μm [20]. Therefore, to accurately validate the effectiveness of BTA to inhibit corrosion in actual copper pipes, it is necessary to study the effect of BTA on copper with carbonaceous residues.

This study discusses performance of BTA as corrosion inhibitor on carbonaceous film-coated copper in synthetic tap water. Scanning electron microscopy (SEM) with energy dispersive spectroscopy (EDS) was used to observe the carbonaceous film-coated copper surfaces. The electrochemical properties of BTA on the carbonaceous film on the copper surfaces were evaluated using a potentiodynamic polarization test and electrochemical impedance spectroscopy (EIS) test. After immersion in an aqueous solution with and without BTA, X-ray photoelectron spectroscopy (XPS) was used to observe the adsorption of BTA on the carbonaceous film-coated copper surface. A potentiostatic polarization test was conducted to confirm the actual corrosion behavior of carbonaceous film-coated copper in the presence or absence of BTA, and then analysis was performed using SEM and EDS.

## 2. Materials and Methods

### 2.1. Specimens and Solutions Preparation

C12200 copper was used in all electrochemical experiments, and the chemical compositions are given in Table 1.

To clarify the effect of carbonaceous residues on copper, two specimens were used: a polished copper specimen (Specimen 1), and half-coated copper with a carbonaceous film (Specimen 2). For electrochemical experiments and carbon coating, the surfaces of all specimens were polished with 1000-grit silicon carbide (SiC) paper, rinsed with ethanol, and dried with N_2_ gas. To fabricate the specimen 2, 10 μL/cm^2^ drawing lubricant was dropped on 1 cm^2^ copper substrate, and then the copper substrate was heated at 300 °C for 1 h in a furnace. The heating temperature complies with the ASTM B88 standard, which is the soft annealing standard of C12200 copper [21,22]. At this temperature, the fatty ester, a constituent of the drawing lubricant, polymerizes through oxidation [23]. Table 2 presents the chemical composition of the drawing lubricant. A carbonaceous film was formed on the copper substrate during annealing. After annealing, specimen 2 was repolished with 1000-grit SiC paper to remove the half area of carbonaceous film. To assess whether the carbonaceous film is formed well on copper by soft annealing, SEM/EDS (SEM-7800F Prime, JEOL Ltd., Tokyo, Japan) analysis was performed on the specimen. In addition, to clarify the precise cross-sectional structure of carbonaceous film, the carbonaceous film was formed on a brittle material, Si wafer, and the cross-section of the unpolished carbonaceous film was analyzed after being broken by freezing in liquid nitrogen. The prepared specimens are tested in synthetic tap water containing 0 and 540 ppm BTA (>99%, commercially purchased, Samchun Chemical, Seoul, Korea) at 25 °C and pH 7.2. The BTA concentration was determined from the amount of the BTA component of the commercial inhibitor–Fernox^®^ Alphi [24]. Table 3 presents the chemical composition of the synthetic tap water used in testing. NaCl, Mg(OH)_2_, CaCO_3_, and H_2_SO_4_ were used to adjust the chemical composition of the synthetic tap water, and 0.1 M HNO_3_ solution was used to control the pH level.

### 2.2. X-ray Photoelectron Spectroscopy (XPS)

XPS was performed with a commercial ESCA system (Axis Supra^TM^, Kratos, Manchester, UK). The excitation source was Al Kα radiation (photoelectron energy = 1486.6 eV). Survey scan spectra were recorded at pass energy of 160 eV, and high-resolution spectra were recorded at pass energy of 20 eV with an energy step of 0.1 eV. XPS spectra were recorded on specimen 2 during 24-h immersion at the open circuit potential in tap water with and without the addition of 540 ppm BTA. Then, the specimens were rinsed with ethanol and dried. To analyze the adsorption of BTA in the presence of the carbonaceous film on specimen 2, three parts of the specimen (copper part, carbonaceous part, and copper-carbonaceous film boundary part) were measured according to the presence or absence of BTA, as shown in Appendix A.

Spectra were deconvoluted as Cu, N, and Cl. Cu was analyzed as both Auger spectra and XPS spectra for more accurate analysis. In Auger[Cu(L_3_M_4,5_M_4,5_)] spectra, the major Cu Auger peak is observed at the binding energy, *E_b_*, of 568.2 eV, and that of Cu_2_O is observed at 570 eV [25,26,27,28]. In Cu 2p_3/2_ XPS spectra, the Cu2p_3/2_ peak of Cu is observed at 932.7 eV and that of Cu_2_O is observed at 932.5 eV [29,30,31,32]. In addition, the peak in N 1s, which is a component of BTA, is observed between 397.9 and 401 eV [25]. The Cl 2p peak of chlorine in CuCl is observed at 198.0 eV [32,33,34].

### 2.3. Electrochemical Tests

The corrosion properties of the specimens were evaluated using a potentiodynamic polarization test and EIS test. All electrochemical experiments were conducted after 24-h immersion using VSP 300 (Bio-Logic SAS, Seyssinet-Pariset, France). To conduct the potentiodynamic polarization test and EIS test, a three-electrode system consisting of two specimens (specimen 1 and specimen 2) as the working electrodes (WE), two pure graphite rods as the counter electrodes (CE), and a saturated calomel electrode (SCE) with a Luggin capillary as the reference electrode (RE) was used. The potentiodynamic polarization test based on the presence or absence of the BTA inhibitor was conducted at a potential sweep of 0.166 mV/s from −250 mV vs. open-circuit potential (OCP) to 1600 mV_SCE_. The EIS test was conducted with an amplitude of 20 mV and a frequency of 100 kHz to 1 mHz. Impedance plots were analyzed on the basis of an equivalent circuit through the ZsimpWin program (ZsimpWin 3.20, Echem Software, Warminster, PA, USA) using the appropriate fitting procedure.

### 2.4. Surface Analysis after Potentiostatic Polarization Test

To investigate the effect of BTA on the carbonaceous film-coated copper, the surface morphology and cross-section analyses were performed by SEM/EDS (SEM-7800F Prime, JEOL Ltd., Tokyo, Japan) after a potentiostatic polarization test. The potentiostatic polarization test was conducted using specimen 2 and at a constant potential of 300 mV_SCE_. The total coulombic charge was 0.0033 mAh and 1.52 mAh, respectively, in the presence or absence of BTA. In the presence of BTA, a Cu−BTA passive film forms and the corrosion rate differs from that of bare copper, so the coulombic charge also differs [7]. The total amount of coulombic charge was obtained by calculating the amount of charge when accelerated for 6 months with each corrosion current density.

## 3. Results and Discussion

### 3.1. Carbonaceous Film Coating Analysis

The cross-section of the copper-coated with the carbonaceous film was analyzed using SEM/EDS. Figure 2a is the cross-sectional image of the copper part, and Figure 2b is the cross-sectional image of the carbonaceous film part on the copper. The carbonaceous film formed on the copper surface has a thickness of 20 μm. EDS data indicate that the carbonaceous film is composed of C and O in the drawing lubricant components (Figure 2c). Appendix A is a cross-sectional electron probe micro analyzer (EPMA) image which was observed near pitting corrosion of a copper pipe for a sprinkler used for 11 years. Appendix A shows that the carbonaceous film is formed with a thickness of 10–20 μm. Appendix A shows that the carbonaceous film has the same composition as drawing lubricant components, which is C and O. This means that carbonaceous residues are present on copper pipes in real field and consistent with the carbonaceous film formed on the specimens in the experiments.

In Figure 3, the carbonaceous film is not completely adsorbed on the substrate and has a granular form and a porous structure with hollow parts. This may cause localized corrosion by crevice corrosion at the boundaries and defects of the carbonaceous film.

### 3.2. XPS Analysis

Figure 4 shows the XPS analysis of specimen 2 after immersed in synthetic tap water for 24 h with and without the addition of BTA 540 ppm. Based on the existence of the BTA, each analysis was conducted on the three parts in specimen 2 to confirm the change in the copper corrosion surface layer due to the carbonaceous film. Since XPS analyses were performed with different specimens under different conditions, only qualitative analyses were performed. Regardless of the presence or absence of BTA, the first peak was observed in the range of 569.4–569.8 eV in the Cu LMM Auger spectra at all three parts. This corresponds to the results in a previous study, where the center of the Cu LMM Auger peak was 568.2 eV, the center of the Cu LMM Auger peak for CuO was 569.2 eV, and the center of the Cu LMM Auger peak for Cu_2_O was 570 eV [25,26,27]. W Liu et al. suggested that the Cu LMM Auger peak in the range of 569–570 eV was due to the formation of CuO/Cu_2_O [26]. Appendix A shows the Cu LMM Auger spectra and deconvoluted results for the copper surface in the presence and absence of BTA. From the results, it can be inferred that the specimen surface is composed of Cu, CuO and Cu_2_O, regardless of the presence or absence of BTA. Similarly, in the Cu 2p_3/2_ XPS spectra, the first peak was observed at 932.2 eV, whereas the Cu 2p_3/2_ peak of Cu_2_O was observed at 932.5 eV, and the Cu 2p_3/2_ peak of Cu was observed at 932.7 eV [29,30,31]. Cu and Cu_2_O appear to exist on copper surfaces regardless of the existence of the BTA. The observation of copper peaks in the carbonaceous parts may be due to the oxidation of copper through the porous structure of the carbonaceous film.

The second peak was observed at 571.8 eV in the Cu LMM Auger spectra in the three parts of specimen 2 with BTA. Similarly, a second peak was observed at 934.4 eV in the Cu 2p_3/2_ XPS spectra. This is consistent with the observations in previous studies, where the Cu(I)-BTA peak was observed at 571.8 eV in the Cu LMM Auger spectra and 934.8 eV in the Cu 2p_3/2_ XPS spectra [25,30]. This peak was not observed in specimen 2 without BTA. In addition, N 1s XPS peak, which is a component of BTA, was observed at 399.5 eV in three parts of specimen 2 in the presence of BTA [32]. This means that that Cu−BTA complex is well formed on all parts in the specimen with carbonaceous film.

The Cl 2p XPS spectra were analyzed to confirm localized corrosion caused by the carbonaceous film. Unlike other parts, 198 eV of the Cl 2p peak was detected in the boundary parts of specimen 2 [32]. Cl 2p peak results that the generation of CuCl and CuCl_2_ increased at the boundary of the carbonaceous film. It is known that the formation of CuCl and CuCl_2_ increases at the Cl concentration increasing site, causing localized corrosion [4]. That is, it was estimated that the formation of CuCl and CuCl_2_ increased with the increase in the Cl concentration in the boundary part.

### 3.3. Electrochemical Analysis

#### 3.3.1. Potentiodynamic Polarization Measurements

Figure 5 shows the potentiodynamic polarization curves for specimen 1 and specimen 2 in synthetic tap water at 25 °C in the presence and absence of BTA. When BTA was added to the solutions, a more noble corrosion potential and the lower corrosion current density were observed. This could be due to the Cu–BTA complex formation, as BTA was adsorbed on the copper surface. Generally, the Cu–BTA complex formation acts as a mixed-type inhibitor to retard the oxidation of copper and reduction in oxygen [7,9]. The breakdown potential (*E_b_*) seen in the BTA-added specimens indicated that the Cu−BTA complex has an effective passive property [9]. Notably, the Cu−BTA complex acts more effectively as an anodic corrosion inhibitor [32]. In Figure 5, the apparent increase in the anodic Tafel slope value and shift in corrosion potential (*E_corr_*) to the noble direction is due to the more effective reduction in the anodic reaction than in the cathodic reaction.

Table 4 lists the electrochemical parameters that were obtained from polarization curves on specimen 1 and specimen 2 in synthetic tap water at 25 °C with the BTA presence and absence. Regardless of the presence or absence of BTA, as shown in Figure 5 and Table 4, specimen 2 had a lower corrosion current density (*i_corr_*) and similar or higher *E_corr_* compared to specimen 1. Raman and Zhang reported that *i_corr_* is decreased and *E_corr_* adjusted more toward a positive direction when metal is coated with a graphene or carbon layer [35,36]. They suggested that the carbon layer or graphene reduced the dissolution of metal, thereby decreasing the corrosion rate by 2–3 times, and increased corrosion resistance, thereby moving *E_corr_* toward a more positive direction. In specimen 2, only half of the area was coated with the carbonaceous film, so it can be expected that *i_corr_* is reduced to half value. The *i_corr_* of specimen 2 was approximately half relatively to that of specimen 1 regardless of the existence of BTA in the solution.

Inhibiting efficiency (*IE%*) can be calculated using the following Equation (1) [9,32];
(1)$$IE\%=100\times [\frac{\left(i_{corr}^{0}-i_{corr}\right)}{i_{corr}^{0}}\;]$$
where $i_{corr}^{0}$ and $i_{corr}$ are the corrosion current densities in the absence and presence of the inhibitor in the solution, respectively. Specimen 1 and specimen 2 have high corrosion *IE%* at 99.83% and 99.78%. It is clear that copper with the carbonaceous film showed a similar BTA adsorption rate and stability as bare copper.

#### 3.3.2. Electrochemical Impedance Spectroscopy (EIS)

Figure 6 shows the Nyquist plots that were obtained for specimen 1 and specimen 2 in synthetic tap water in the presence and absence of BTA. When BTA is added, it is shown in Figure 6 that both specimen 1 and specimen 2 increased the capacitive loop as well as corrosion inhibition efficiency. Figure 6 also shows the equivalent circuit for copper based on the presence and absence of BTA [29,32,37]. The proposed equivalent circuit fits well with EIS data. The equivalent circuit comprised the following elements. *R_s_* is the solution resistance. *R1* is the resistance due to defects in the layer formed over copper or the formation of ionic conduction paths through pores. In the absence of BTA, resistance is due to copper oxide, and in the presence of BTA, resistance is due to Cu−BTA complex formation [29,32,37]. *CPE1* is the barrier capacitance corresponding to *R1*, *R_ct_* is the charge transfer resistance, and *CPE2* is the capacitance due to the electric double layer generated at the interface between the rust/ barrier and metal substrates. The constant phase element (*CPE*) is a non-ideal capacitance with a varying *n*, expressed by Equation (2) [9,25,36,38]
(2)$$Q=Z_{CPE}=\frac{1}{Y_{0}{\left(j\omega \right)}^{n}}$$
where *Y*_0_ is the magnitude of the *CPE*, and *n* is a parameter by frequency dispersion. The parameter *n* is affected by non-homogeneity and surface roughness [9]. When *n* = 1, *Q* represents the ideal capacitor *C*, and when *n* = 0, *Q* becomes a simple resistor. Parameter *n* generally has a value of 0.5–1, and *n* = 0.5 indicates a process in which copper ions are diffused through the pores of the oxide layer [9,25]. Table 5, which is calculated from EIS data, shows that the value of *n*_1_ is close to 0.5 for both specimen 1 and specimen 2 in the absence of BTA, and the value of *n*_1_ is close to 1 for both specimen 1 and specimen 2 in the presence of BTA. It can be regarded as the diffusion of copper ions through the defects of a copper oxide layer in the absence of the BTA, and formation of the Cu−BTA complex in the presence of the BTA. The Cu−BTA complex formation acts as an ideal capacitor and a barrier.

The total amount of resistance and inhibitor efficiency (*IE%*) are calculated using Equations (3) and (4) [9]
(3)$$R_{total}=R1+R2$$
(4)$$IE\%=100\times \left(\frac{R_{total}-R_{total}^{0}}{R_{total}}\right)$$

In Equation (3), *R_total_* has a direct correlation with corrosion resistance [35]. In Equation (4), *R_total_* and *R^0^_total_* are total resistance in the presence and absence of BTA, respectively. Figure 7a shows the *R_total_* value based on the presence or absence of BTA in specimen 1 and specimen 2. Both the *R_total_* and *R^0^_total_* of specimen 2 increased by approximately 2 times compared to the resistances of specimen 1. Half of the specimen 2 was coated with a carbonaceous film. It can be expected that the area coated with a carbonaceous film of specimen 2 has high resistance, so that, only the area uncoated with the carbonaceous film is measured [35,36]. Therefore, the resistance of specimen 2 is doubled compared to specimen 1 because only half area of the specimen 1 is measured. Figure 7b shows the *IE%* obtained from PD and EIS for copper based on the presence and absence of BTA. The *IE%* values obtained from PD and EIS show a tendency to slightly decrease the efficiency in specimen 2. This is presumed to be due to the changes in the Cu−BTA complex formation by the carbonaceous film of specimen 2. According to the XPS results, the Cl concentration increased at the boundary of the carbonaceous film of specimen 2. The alteration in the corrosive environment at the boundary may change the mechanism of the Cu−BTA complex formation. However, the *IE%* of specimen 2 from PD and EIS data shows very high efficiencies of 99.78 and 99.91%, respectively. Therefore, BTA corrosion inhibitor is sufficient to suppress corrosion in copper covered with the carbonaceous film.

### 3.4. Surface Analysis after Potentiostatic Polarization Test

Figure 8 shows the results of the element mapping analysis of the specimen 2 after potentiostatic polarization test as the BTA presence and absence. Uniform corrosion occurred in the copper part or carbonaceous film part of the specimen 2 when BTA was not added, whereas the corrosion behavior was different in the boundary part of the specimen 2. Figure 8a,b shows EDS mapping analyses of the boundary part of the surface section of the specimen 2 based on the absence and presence of BTA, respectively. Under both conditions, C and O, constituents of the drawing lubricant, were detected in the carbonaceous film part. Cu and O were detected in the copper part. This is because Cu_2_O was formed in the copper part [29]. However, in Figure 8b, only a little of the O element is detected in the Cu part compared to that shown in Figure 8a. This is because the formation of Cu_2_O was restricted due to the formation of the Cu−BTA complex [7,32].

Depending on the presence and absence of BTA, EDS mapping analysis is utilized to exhibit the distribution of Cl at the boundary part of specimen 2. In Figure 8a, a large number of Cl elements are detected at the boundary part of specimen 2. As Cu was also detected in the Cl region, it appears to be due to the formation of CuCl and CuCl_2_. However, in Figure 8b, the Cl element is hardly detected at the boundary of specimen 2. In Figure 8c, which is a condition in the absence of BTA, localized corrosion occurred at the boundary of the carbonaceous film, and the Cl element was concentrated in that part. However, in Figure 8d, which is a condition in the presence of BTA, localized corrosion did not occur at the boundary of the carbonaceous film, and a Cl element was not concentrated. When BTA is added, it appears that the formation of the Cu−BTA complex limits the formation of CuCl and CuCl_2_ and prevents the concentration of Cl. The related corrosion mechanisms are proposed below.

Several researchers have studied the corrosion behavior of copper. The corrosion behavior of copper varies depending on the Cl^-^ concentration in an aqueous solution [39]. Copper reacts according to Equations (5)–(8) as anodic reactions depending on the Cl concentration in an aqueous solution [4].
(5)$$\mathrm{Cu}\to {\mathrm{Cu}}^{+}+e^{-}$$
(6)$${\mathrm{Cu}}^{+}+{\mathrm{Cl}}^{-}\to \mathrm{CuCl}$$
(7)$$2\mathrm{CuCl}+H_{2}O\to {\mathrm{Cu}}_{2}O+2H^{+}+2{\mathrm{Cl}}^{-}$$
(8)$$\mathrm{CuCl}+{\mathrm{Cl}}^{-}\to {\mathrm{CuCl}}_{2}^{-}$$

Equation (7) occurs when the Cl^−^ ion concentration is relatively low (NaCl electrolytes containing Cl^−^ concentration <10^−3^ M), and Equation (8) occurs when Cl^−^ ion concentration is relatively high (NaCl electrolytes containing Cl^−^ concentration >10^−2^ M). The Cl^−^ ion concentration of synthetic tap water is 15.8 ppm, and it is expected that Cu_2_O will be formed by the reaction of Equation (7) on the entire copper surface. It can be seen that Cu_2_O was formed at the copper part in Figure 8a. However, Cu_2_O was not formed and Cl was concentrated at the boundary part of the carbonaceous film-copper. The carbonaceous film acts as an efficient cathode, and the small gap between carbonaceous film and substrate acts as an anodic site, which accelerates corrosion due to the large cathode–small anode effect [17]. In the same way as in the boundary part, galvanic corrosion occurs between the carbonaceous film and copper, accelerating the corrosion of copper in the boundary part. In addition, as shown in Figure 3, the carbonaceous film does not completely adsorb on the metal surface, creating a gap, and crevice corrosion may occur in the boundary part. Figure 8c shows that corrosion occurs under the carbonaceous film in the boundary part. When the corrosion occurs in this way, the diffusion of ions in the gap is limited, the inside of the gap is acidified, and high Cl^−^ ion concentration conditions are formed. Corrosion is further accelerated by forming an oxygen-concentration cell with the surrounding area, and Equation (8) occurs inside the gap to produce CuCl and ${\mathrm{CuCl}}_{2}^{-}$ instead of Cu_2_O, a protective oxide layer, resulting in a more corrosive environment [40,41]. At this time, oxygen reduction reaction occurs at the surrounding area according to Equation (9).
(9)$$O_{2}+2H_{2}O+4e^{-}\to 4{\mathrm{OH}}^{-}$$

Figure 9a shows a schematic diagram of the localized corrosion mechanism at the boundary part without BTA.

When BTA is added to the solution, the concentration of Cl in the boundary part disappears and corrosion acceleration does not occur, as shown in Figure 8b,d. Figure 9b is a schematic diagram of the corrosion prevention mechanism at the boundary part with BTA. When BTA is added to the solution, a Cu−BTA complex is formed at the boundary of the carbonaceous film to prevent localized corrosion. Many researchers have reported that Cu−BTA protective layer was formed by BTA even in solutions containing CuCl and ${\mathrm{CuCl}}_{2}^{-}$ [7]. Modestov et al. proposed that BTA acts in a chloride solution according to Equation (10) [42]. That is, BTA alleviates localized corrosion by forming a Cu−BTA complex at the anodic site at the boundary of the carbonaceous film.
(10)$${\mathrm{CuCl}}_{2}^{-}+\mathrm{BTAH}\to \mathrm{Cu}-\mathrm{BTA}+2{\mathrm{Cl}}^{-}+H^{+}$$

Furthermore, when BTA is added to the solution, a Cu−BTA complex is formed in the copper part to mitigate the effect of the oxygen-concentration cell. BTA forms Cu−BTA complex by Equation (11) in copper parts [7,9,32].
(11)$$n{\left(\mathrm{BTAH}\right)}_{\mathrm{ads}}+\mathrm{nCu}\to {\left(\mathrm{Cu}-\mathrm{BTA}\right)}_{n}+{\mathrm{nH}}^{+}+{\mathrm{ne}}^{-}$$

BTA is a mixed-type inhibitor and suppressed the oxygen reduction reaction on the surface of Cu−BTA. That is, BTA mitigates localized corrosion by inhibiting the formation of the oxygen-concentration cell at the boundary of the carbonaceous film.

## 4. Conclusions

The corrosion inhibition effect of BTA on the copper surface with a carbonaceous film in synthetic tap water was investigated using XPS, potentiodynamic polarization, EIS, potentiostatic polarization, and SEM/EDS. According to experimental results, the following conclusions were drawn:In solutions containing BTA, Cu−BTA complex and N elements were detected in the copper part, carbonaceous film-copper boundary part, and carbonaceous film part through XPS analysis. In addition, BTA decreased *i_corr_* and increased *R1* and *n_1_*. This is because BTA adsorbs well on the entire surface of copper with carbonaceous film to form a Cu−BTA protective layer;XPS and SEM/EDS analyses indicated that localized corrosion and Cl concentration occurred in the carbonaceous film–copper boundary. It was inferred that this was due to crevice corrosion caused by the gap between the carbonaceous film and copper surface and the galvanic corrosion between the carbonaceous film and copper;BTA mitigates localized corrosion at the anodic site by forming the Cu−BTA complex in the carbonaceous film–copper boundary part. In addition, the formation of a Cu−BTA complex in the copper part inhibits the formation of an oxygen-concentration cell. Consequently, BTA suppresses localized corrosion caused by the carbonaceous film.

## Figures and Tables

**Figure 1 materials-14-02722-f001:**
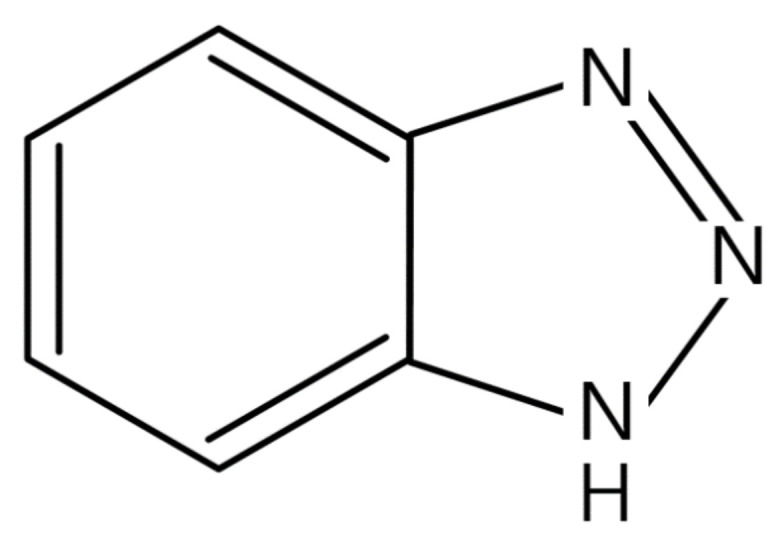
Molecular structure of benzotriazole.

**Figure 2 materials-14-02722-f002:**
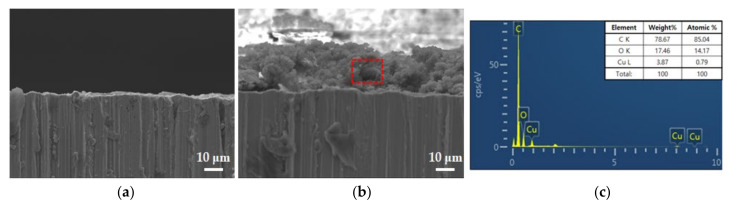
Cross-sectional SEM images of the specimens; (**a**) SEM images of specimen 1 (1000×); (**b**) SEM images of the carbonaceous film on specimen 2 (1000×); (**c**) EDS result of the carbonaceous film (red dotted-box).

**Figure 3 materials-14-02722-f003:**
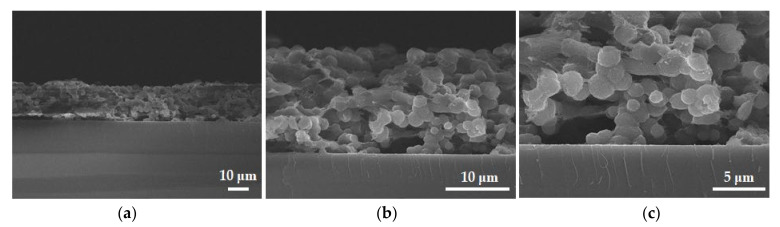
Cross-sectional SEM images of carbonaceous film on Si wafer; (**a**) 1000×; (**b**) 3000×; (**c**) 5000×.

**Figure 4 materials-14-02722-f004:**
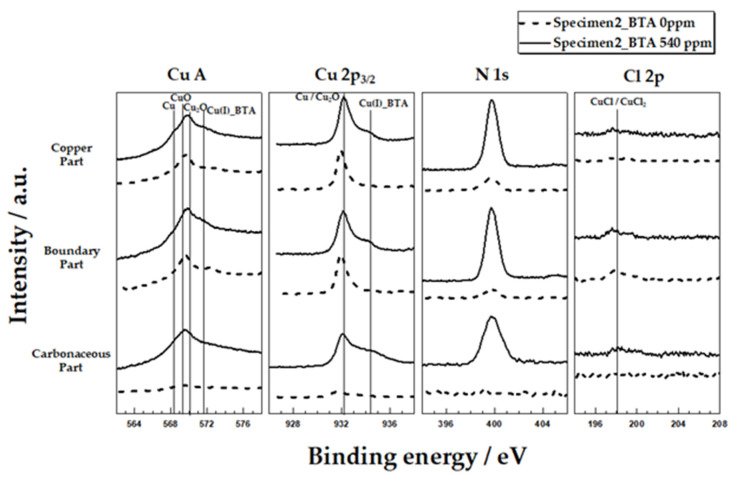
Cu LMM X-ray induced spectra and Cu 2p3/2, N 1s, and Cl 2p XPS spectra recorded at the surface of copper after 24-hour immersion in synthetic tap water at 25 °C based on the presence and absence of BTA.

**Figure 5 materials-14-02722-f005:**
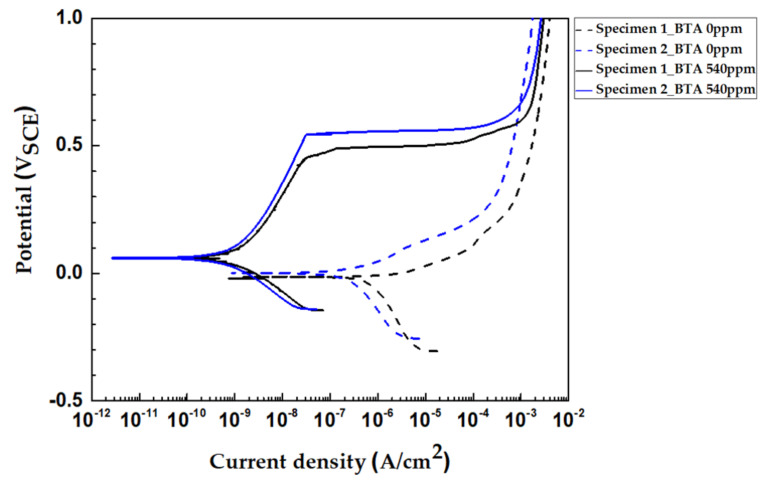
Potentiodynamic polarization curves for specimen 1 and specimen 2 in synthetic tap water at 25 °C according to the presence and absence of BTA.

**Figure 6 materials-14-02722-f006:**
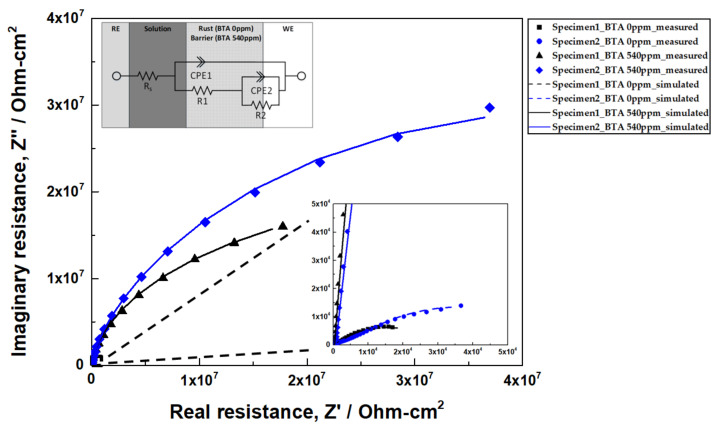
Nyquist plots for specimen 1 and specimen 2 and equivalent circuit for fitting in synthetic tap water at 25 °C based on the presence and absence of BTA.

**Figure 7 materials-14-02722-f007:**
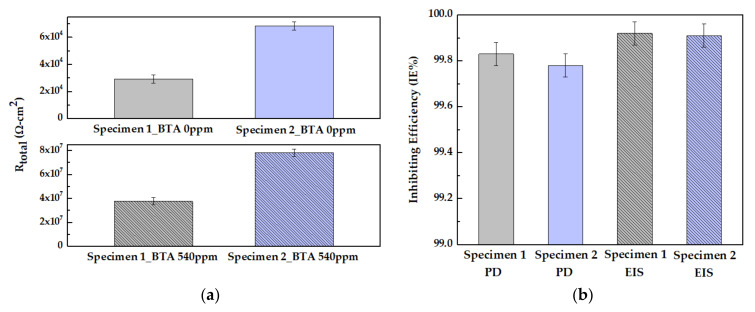
(**a**) The value of the total resistance of copper as a function of BTA; (**b**) The inhibiting efficiency (*IE%*) of copper as a function of BTA from potentiodynamic (PD) polarization curve and EIS.

**Figure 8 materials-14-02722-f008:**
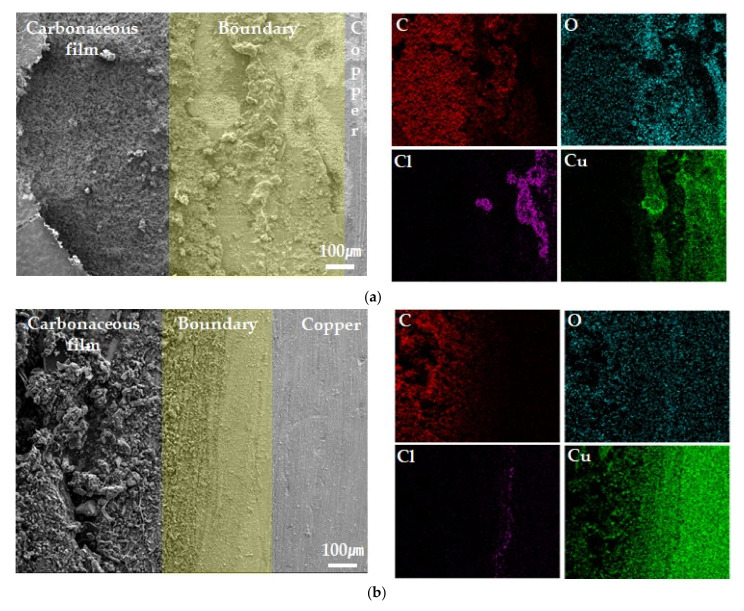
EDS mapping analysis of the copper-carbonaceous film boundary parts of specimen 2 after potentiostatic polarization test based on the presence or absence of BTA; (**a**) surface analysis of specimen 2 without BTA; (**b**) surface analysis of specimen 2 with BTA; (**c**) cross-section analysis of specimen 2 without BTA; (**d**) cross-section analysis of specimen 2 with BTA.

**Figure 9 materials-14-02722-f009:**
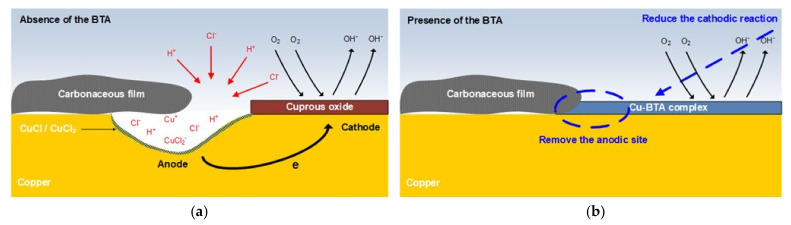
(**a**) Schematic diagram of localzed corrosion mechanism at the boundary part of the carbonaceous film of copper in the absence of BTA; (**b**) schematic diagram of mechanism to prevent localized corrosion at the boundary part of the carbonaceous film of copper in the presence of BTA.

**Table 1 materials-14-02722-t001:** Chemical composition of C12200 copper (wt.%).

Elements	Composition
Copper	99.9
Phosphorus	0.015–0.04
Carbon	0004
Silicon	0.01
Fe	0.01

**Table 2 materials-14-02722-t002:** Chemical composition of drawing lubricant (wt.%).

Elements	Composition
Polyisobutylene (PIB)	80–85
Fatty ester	15–20

**Table 3 materials-14-02722-t003:** Chemical composition of synthetic tap water (ppm) used in testing.

pH	Cl^−^	Mg^2+^	Ca^2+^	SO_4_^2−^	BTA
7.2	15.8	12.9	51.7	13.2	0, 540

**Table 4 materials-14-02722-t004:** Electrochemical parameters from polarization measurements on specimen 1 and specimen 2 in synthetic tap water at 25 °C based on the presence and absence of BTA.

Specimen	*C_BTA_* (ppm)	*E_corr_* (mV_SCE_)	*i_corr_* (nA/cm^2^)	*E_b_* (mV_SCE_)	*IE%*
Specimen 1	0	−17.83	759.95	-	-
	540	60.27	1.32	451.89	99.83
Specimen 2	0	−2.83	351.51	-	-
	540	56.87	0.76	544.14	99.78

**Table 5 materials-14-02722-t005:** Electrochemical parameters from EIS measurements on specimen 1 and specimen 2 in synthetic tap water at 25 °C based on the presence and absence of BTA.

Specimen	*C_BTA_* (ppm)	*R_s_* (Ω-cm^2^)	*R1* (Ω-cm^2^)	*CPE1* (F/cm^2^)	*n_1_*	*R2* (Ω-cm^2^)	*CPE2* (F/cm^2^)	*n_2_*	*IE%*
Specimen 1	0	2.49 × 10^2^	1.86 × 10^3^	2.22 × 10^−5^	0.66	2.72 × 10^4^	9.00 × 10^−5^	0.51	-
	540	4.35 × 10^2^	8.14 × 10^6^	1.51 × 10^−6^	0.96	2.94 × 10^7^	4.59 × 10^−7^	0.66	99.92
Specimen 2	0	6.78 × 10^2^	7.36 × 10^3^	4.74 × 10^−5^	0.49	6.03 × 10^4^	1.59 × 10^−4^	0.51	-
	540	8.15 × 10^2^	1.07 × 10^7^	6.01 × 10^−7^	0.95	6.75 × 10^7^	1.54 × 10^−7^	0.58	99.91

## Data Availability

Not applicable.

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
