# Peer review of "Effect of Benzotriazole on the Localized Corrosion of Copper Covered with Carbonaceous Residue"

_materials, 2021, doi:10.3390/ma14112722_

Round 1
Reviewer 1 Report
- In Fig.2, the caption “carbonaceous film on Si wafer”, so why do you use the Si wafer?
- In Fig.3, lines related to the BTA 0 ppm samples are not clear enough.
- In Fig.3, the peaks of Cu LMM is not at the line of Cu2O, what is the reason? is there any other component?
- In Fig.5, is there any problem about the Y-axis? Since the Y-axis of the insert is not consistent with the Y-axis of the main diagram.
- In Fig.7, (a) and (c) is related to the specimen without BTA, it can be seen from (a) that Cl is mainly distributed on the Copper region, while in (c), Cl is mainly distributed on the boundary region. Why?
Author Response
Thank you for your comment and it will improve the paper. As you mentioned, the paper was revised.
① To clarify the precise cross-sectional structure of the carbonaceous film, a brittle material, Si wafer, was used. For copper, the analysis of the unpolished cross-sectional structure of the carbonaceous film is limited due to its high ductility, even if it breaks after freezing in liquid nitrogen.
② The line of BTA 0 is clearly expressed by making the line thicker as shown in the attachment.
③ The first peak you mentioned is considered to be the overlapping of Cu, CuO, and Cu2O peaks. The first peak was observed in the range of 569.4 eV to 569.9 eV in the Cu LMM Auger spectra at all three parts. W Liu et al. suggested that the CuO LMM Auger peak is 569.2 eV [33]. As a result of deconvoluted Cu LMM Auger spectra, the first peak was composed of Cu, CuO, and Cu2O. In addition, W Liu et al. suggested that the Cu LMM Auger peak in the range of 569-570 eV is due to the formation of CuO/Cu2O [33].
The deconvoluted results are shown in the attachment.
④ There is a mistake about magnification part boundary. The magnification part is changed as shown in the attachment.
⑤ In Figure 7 (a), there is a mistake in marking the boundary area by focusing on Cl in the center of the carbonaceous film. Cl present in the copper part is actually the boundary part. This is because C exists right next to the Cl presence region of the copper part you pointed out. To avoid misunderstanding, Figure 7 (a)’s boundary area is corrected as shown in the attachment.
The reason Cl is mainly distributed on the boundary region is thought to be due to galvanic corrosion between the carbonaceous film and the copper and crevice corrosion in the gap between the carbonaceous film and the copper. It is believed that corrosion is accelerated by galvanic corrosion and crevice corrosion, creating a high Cl concentration environment to form CuCl2 instead of Cu2O.
Reference
- Liu, W.; Wang, B.; Cui, C.; Zhang, Y.; Wang, L.; Wang, Z., The surface restructuring of copper oxides with mixed oxidation-states and their efficient CO oxidation properties. Materials Letters 2021, 289, 129378.

Reviewer 2 Report
This research work presents results related to corrosion inhibition effect of Benzotriazole on Copper surface. In general, the results are well presented and written in a clear way. The characterization techniques used in this work to show the effect of above-mentioned corrosion inhibitor on corrosion resistance of copper are also adequate. I do not have any major revision comments. However, I would like to recommend that in the introduction part, Benzotriazole's structure & working mechanism as a corrosion inhibitor should be mentioned. Other than this, I have some minor comments which are present in the attached file.

Author Response
Thank you for your comment and it will improve the paper. As you mentioned, the paper was revised.
1. The introduction part was revised.
- Benzotriazole’s structure & working mechanism was mentioned.
2. I corrected a manuscript according to the minor comments.
Reviewer 3 Report
The manuscript “Effect of Benzotriazole on the Localized Corrosion of Copper Covered with Carbonaceous Residue” by Lee et al., presents a comparative study of the effect of benzotriazole (BTA) on copper in the presence of carbonaceous film in synthetic tap water. The authors have utilized scanning electron microscopy, elemental analysis, as well as electrochemical measurements to assess the protection efficiency of BTA. The study is well conducted and the text is relatively clearly written. Although a vast number of such studies can be already found in the literature, there are some novel aspects in this manuscript that make it suitable for publication. The authors, however, may want to consider the following minor points to improve the quality of their manuscript.
Lines 25-29: This sentence is ambiguous and hard to read.
Line 41: “inhibitory effectiveness” can be replaced by “inhibition efficiency”
Line 41: “In addition, similar research has been conducted in different conditions [7, 12-15].” Such sentences are not suitable for a scientific publication, as they do not provide any useful information to the readers.
Line 57: “research” can be replaced by “assess” or “study”.
Lines 79-81: “To fabricate ….” is grammatically incorrect. Please re-write.
Line 93: “respectively” does not fit here.
Lines 93-94: “The BTA concentration …” please re-write this sentence.
Lines 121-125: As it is written, it sounds like two specimens were simultaneously used in the electrochemical measurements. Please re-phrase.
Line 137: please replace “based on the” with “in the”
Lines 138-140: Since this is from literature, it should read “ In the presence of …. passive film forms and the corrosion rate differs from … so that the Coulombic charge also differs”
Lines 140-141: This sentence is unclear.
Line 196: “P” in “polarization” should not be capitalized.
Line 198: “as the BTA presence and absence” should read “in the presence and absence of BTA”
Line 210: “Regardless of the BTA containing” should be re-written
Line 233: “as the BTA presence and absence” should read “in the presence and absence of BTA”
Line 234: should be re-written (a passive tense may be used here).
Figure 5: It would be useful to see the experimental EIS results overlapped with the fitted curves, to confirm the accuracy of the proposed model and obtained fit parameters.
Author Response
Thank you for your comment and it will improve the paper. As you mentioned, the paper was revised.
- I corrected a manuscript according to the minor comments.
- The Figure 5 has been corrected by adding fitted curves as shown in the attachement.

Reviewer 4 Report
There are the following questions about the article:
- In Figure 3, the intensity of the Cu LMM peak in Copper and Boundary Parts on a sample without BTA is less than on a sample with BTA. Shouldn't it be the other way around? Since BTA inhibits the anodic dissolution of copper, there should be less of it.
- From Fig. 7d it follows that C, Cl, and O are evenly distributed in the copper alloy. Is this a measurement error? In Figure S2, there is no such phenomenon.
There are the following comments and recommendations:
- Keywords do not reflect the essence of the work, they need to be changed. The following keywords can be suggested: copper corrosion, local corrosion, BTA inhibitor, inhibition, carbonaceous residue.
- The rate of copper corrosion could be estimated using a depth corrosion indicator, since there is a value of the corrosion current. This will allow you to estimate the service life of copper pipes.
Author Response
Thank you for your comment and it will improve the paper. As you mentioned, the paper was revised.
1. BTA inhibits the dissolution of the copper anode, so I agree with the comment that the intensity of the peak should be small. However, it is possible to quantitatively analyze the same specimen by XPS analysis, but the quantitative analysis is not accurate for other specimens. The XPS analysis conducted in this study was performed under different conditions for different specimens. It can be changed according to the analysis time, angle of emission of photoelectron, and the overlap of the Cu-BTA peak. Therefore, in Figure. 3, it is better to consider only the qualitative analysis according to the presence or absence of a peak. To avoid misunderstanding, the following sentence was added.
--> Since XPS analyses were performed with different specimens under different conditions, only qualitative analyses were performed.
2. In Figure 7(d), C, Cl, and O are evenly distributed because contaminants are detected. The table in the attachment shows the EDS mapping results in Figure 7(d). In fact, the Cl atomic % is 0, which can be seen undetected. Similarly, in the EPMA analysis in Figure S2, it is consistent that Cl and O were detected in blue in the copper region. That is, Figure 7(d) and Figure S2 can be seen as the same phenomenon.
There are the following comments and recommendations:
1. The keywords are revised according to your advice.
2. Please check the attachment for answers to this comment.

Round 2
Reviewer 4 Report
-